

# The addition of very light loads into the routine testing of the bench press increases the reliability of the force–velocity relationship

Jesualdo Cuevas-Aburto[1], David Ulloa-Díaz[1], Paola Barboza-González[2], Luis Javier Chirosa-Ríos[3] and Amador García-Ramos[1,3]

[1] Departamento de Ciencias del Deporte y Acondicionamiento Físico, CIEDE, Universidad Católica de la Santísima Concepción, Concepción, Chile
[2] Facultad de Educación y Ciencias Sociales, Universidad Andres Bello, Concepción, Chile
[3] Departamento de Educación Física y Deportiva, Facultad de Ciencias del Deporte, Universidad de Granada, Granada, España

Corresponding author
Amador García-Ramos,
amagr@ugr.es

## ABSTRACT

**Background:** The aim of this study was to examine whether the addition of very light loads for modeling the force–velocity (F–V) relationship during the bench press (BP) exercise can confirm its experimental linearity as well as to increase the reliability and concurrent validity of the F–V relationship parameters (maximum force ($F_0$), maximum velocity ($V_0$), F–V slope, and maximum power ($P_{max}$)).

**Method:** The F–V relationship of 19 healthy men were determined using three different methods: (I) 6-loads free method: six loads performed during the traditional free-weight BP exercise ($\approx$ 1–8–29–39–49–59 kg), (II) 4-loads free method: four loads performed during the traditional free-weight BP exercise ($\approx$ 29–39–49–59 kg), and (III) 4-loads Smith method: four loads performed during the ballistic bench press throw exercise in a Smith machine ($\approx$ 29–39–49–59 kg).

**Results:** The linearity of the F–V relationship was very high and comparable for the three F–V methods ($p = 0.204$; median Pearson's correlation coefficient ($r$) = 0.99). The three methods were ranked from the most to the least reliable as follows: 6-loads free (coefficient of variation (CV) range = 3.6–6.7%) > 4-loads Smith (CV range = 4.6–12.4%) > 4-loads free (CV range = 3.8–14.5%). The higher reliability of the 6-loads free method was especially pronounced for F–V slope ($CV_{ratio} \geq 1.85$) and $V_0$ ($CV_{ratio} \geq 1.49$) parameters, while the lowest difference in reliability was observed for $F_0$ ($CV_{ratio} \leq 1.27$). The 6-loads free and 4-loads free methods showed a very high concurrent validity respect to the 4-loads Smith method for $F_0$ and $P_{max}$ ($r \geq 0.89$), a moderate validity for the F–V slope ($r = 0.66–0.82$), and a low validity for $V_0$ ($r \leq 0.37$).

**Discussion:** The routine testing of the F–V relationship of upper-body muscles through the BP exercise should include trials with very light loading conditions to enhance the reliability of the F–V relationship.

## INTRODUCTION

The force–velocity–power (F–V–P) relationships are being increasingly used to explore the function of the neuromuscular system (*Alcazar et al., 2018*; *Cross et al., 2018*; *García-Ramos et al., 2018b*). The modeling of the F–V–P relationships in single- and multi-joint tasks allows to determine mechanical limits of the neuromuscular system to produce force ($F_0$; theoretical maximal force capacity at null velocity), velocity ($V_0$; theoretical maximal velocity capacity until which the system is still able to produce force), and power ($P_{max}$) (*Jaric, 2015*). An important characteristic of the F–V relationship when compared to the standard testing procedure conducted under a single mechanical condition (e.g., unloaded jump) is that the outcomes of the F–V relationship are able to distinguish between $F_0$ and $V_0$ capacities (*Jaric, 2015*). Note that while the force and velocity outputs obtained under a single load are inter-dependent (a higher force impulse inevitably produces a higher velocity), $F_0$ and $V_0$ are independent of each other (i.e., a change in $F_0$ can be observed without a change in $V_0$). In this regard, it has been proposed that each subject presents an optimal balance between $F_0$ and $V_0$ capacities and, consequently, the ratio between $F_0$ and $V_0$ (i.e., F–V slope) can be used to implement individualized resistance training programs (*Morin & Samozino, 2016*; *Samozino et al., 2012*). Due to the undeniable importance of the outcomes of the F–V relationship (i.e., $F_0$, $V_0$, F–V slope, and $P_{max}$), a number of studies have been conducted to refine the testing procedure of the F–V relationship in different exercises (*Alcazar et al., 2017*; *García-Ramos et al., 2017*, *2018a*; *Lemaire et al., 2014*; *Pérez-Castilla et al., 2018*).

The standard testing procedure used to assess the F–V relationship consists of recording force and velocity outputs by means of different equipment (e.g., force plate, linear position transducers, accelerometers, phone applications, etc.) under multiple loading conditions (*Cuk et al., 2014*; *García-Ramos et al., 2016*; *Rahmani et al., 2018*). While the number of loads seems to have only trivial effects on the reliability and accuracy of the F–V relationship, it is recommended to maximize the distance between the lightest and the heaviest loads to provide one experimental point close to the velocity-intercept and another experimental point close to the force-intercept (*García-Ramos & Jaric, 2018*; *Pérez-Castilla et al., 2018*). However, especially on the field, the range of loads that can be applied to determine the F–V relationship is limited by the characteristics of the exercise tested (e.g., the minimum load in vertical jumps typically represents the subject's own body mass, which provides a point that is located in the middle of the F–V relationship) (*García-Ramos et al., 2017*; *Samozino et al., 2012*). The most common and often used method for testing the F–V relationship of upper-body muscles consist of recording force and velocity outputs against different loading conditions during the bench press throw (BPT) exercise performed in a Smith machine (*García-Ramos et al., 2016*; *Pérez-Castilla et al., 2018*; *Rahmani et al., 2018*; *Sreckovic et al., 2015*). Since the minimum load imposed by the unloaded Smith machine system is approximately 17–20 kg, the resulting experimental points to determine the F–V relationship during the BPT exercise are generally closer to the force-intercept than to the velocity-intercept (i.e., force-biased) (*García-Ramos et al., 2016*; *Pérez-Castilla et al., 2018*; *Rahmani et al., 2018*;
*Sreckovic et al., 2015*). For example, while in previous studies the difference between the mean force output recorded against the heaviest load and $F_0$ ranged from 19.2% to 24.9%, the difference between the velocity output recorded against the lightest load and $V_0$ ranged from 59.5% to 86.8% (*García-Ramos et al., 2016*; *Sreckovic et al., 2015*).

First, since a narrow range of loads have commonly been used for testing the F–V relationship during the BPT exercise, we do not know whether the F–V relationship would remain linear when very light loads (e.g., one and eight kg) are used for the F–V modeling. Second, it has been suggested that the distance of the experimental points to the axis intercepts influences the reliability of the F–V relationship parameters (the larger the distance, the lower the reliability) (*García-Ramos & Jaric, 2018*; *Pérez-Castilla et al., 2018*). Therefore, it would be important to elucidate whether the use of the data of two very light loads during the bench press (BP) could enhance the reliability of $V_0$ and related F–V relationship parameters (i.e., F–V slope and $P_{max}$). Finally, it would be interesting to assess the concurrent validity of the F–V relationship parameters obtained during the free-weight BP exercise respect to the most common procedure of determining the F–V relationship consisting of BPTs performed in a Smith machine. Note that the main advantage of using the free-weight BP exercise for modeling the F–V relationship is the possibility to obtain the experimental points closer to the velocity-intercept.

The present study was designed to elucidate whether the evaluation of the F–V relationship during the BPT exercise using a machine-guided bar, which is the usual testing machine, could be replaced by a traditional BP evaluation without guided bar since the free-weight BP could provide experimental points closer to the velocity-intercept. Specifically, the objectives of the present study were (I) to elucidate whether the F–V relationship during the BP exercise remains linear when experimental points close to the velocity-intercept are modeled, (II) to compare the reliability of the F–V relationship parameters between the standard BPT testing procedure performed in a Smith machine (4-loads Smith method) and the testing procedure based on the BP performed with a free-weight barbell against the same loads (4-loads free method) as well as adding two light loads of one and eight kg (6-loads free method), and (III) to explore the concurrent validity of the F–V relationship parameters obtained from the 4-loads free and 6-loads free methods with respect to the standard 4-loads Smith method.

We hypothesized that (I) the linearity of the F–V relationship would be high and not significantly different between the three methods, (II) the 6-loads free method would provide $V_0$, F–V slope and $P_{max}$ parameters with higher reliability than the 4-loads Smith and 4-loads free methods, while no significant differences would be observed for $F_0$, and (III) the 6-loads free method would provide more valid F–V relationship parameters than the 4-loads free method.

## METHOD

### Subjects

A total of 19 men (mean ± standard deviation; age: 24.0 ± 3.8 years, height: 1.73 ± 0.07 m, body mass: 80.0 ± 12.2 kg, BP one-repetition maximum (1RM): 95.3 ± 23.3 kg

(1.20 ± 0.27 kg·kg$^{-1}$)) volunteered to participate in this study. Subjects were physically active sports science students ($n = 8$), rugby players ($n = 6$) and weightlifting competitors ($n = 5$). All subjects were experienced in resistance training (3.9 ± 3.4 years) and familiar with the BP exercise. Subjects reported no chronic diseases or recent injuries that could compromise testing. All subjects were informed of the procedures to be utilized and signed a written informed consent form prior to investigation. The study protocol was approved by the Institutional Review Board of the University of Granada (no: 491/CEIH/2018).

## Study design

Subjects attended the laboratory on three occasions with 72–96 h of rest between sessions. The 1RM in the free-weight BP exercise was determined in the first session. The second and third testing sessions were identical and they consisted of the free-weight BP exercise performed against six loading conditions and the BPT exercise performed in a Smith machine against four loading conditions. The three testing sessions were performed at the same time of the day for individual subjects (± 1 h).

## Determination of the 1RM

Before the commencement of the testing procedure for determining the 1RM during the BP exercise, subjects self-selected the position of the grip width, which was kept constant throughout all testing sessions. The warm-up consisted of 5 min of jogging at a self-selected pace and joint mobility exercises of the shoulders, elbows, wrists and hips, followed by two sets of five repetitions of the BP exercise performed with fixed loads of 20 and 30 kg. Initial load of the test was set at 40 kg for all subjects and was progressively increased in 10 kg until the attained mean velocity of the bar was lower than 0.5 m·s$^{-1}$. From that moment, the load was increased in steps of five to one kg until the 1RM load was achieved. The magnitude of the increment was decided by a skilled investigator after reaching a consensus with the subject. Three minute of rest were implemented for light–moderate loads (mean velocity ≥ 0.50 m·s$^{-1}$) and 5 min for heavier loads (mean velocity < 0.50 m·s$^{-1}$). Two repetitions were performed with light–moderate loads and only one repetition was performed with heavier loads.

The concentric-only BP variant was used in the three sessions of the present study (*Pallarés et al., 2014*). The standard five-point body contact position technique (head, upper back, and buttocks firmly on the bench with both feet flat on the floor) was used. Subjects initiated the BP exercise with their elbows fully extended and then they lowered the bar until it contacted with their chest at the level of the nipples. Subjects maintained the bar in contact with their chest for 1 s and then they performed the concentric phase of the exercise at the maximum possible velocity. The BP (i.e., the bar was always in contact with the subjects' hands) was performed with a free-weight barbell, while the BPT (subjects were instructed to throw the bar as high as possible) was performed in a Smith machine. Two spotters were standing on each side of the Smith machine to catch the bar during its descent in the BPT exercise.

## Force–velocity relationship

The general warm-up consisted of 5 min of jogging at a self-selected pace and joint mobility exercises. Thereafter, subjects performed 10, five and two repetitions during the free-weight BP exercise at the 30%1RM, 60%1RM, and 80%1RM, respectively. To complete the warm-up, subjects performed two repetitions of the BPT exercise against 29 kg. The first testing protocol started 4 min after completing the warm-up.

The testing protocol of both testing sessions consisted of the BP performed against six loading conditions and the BPT performed against four loading conditions. Two repetitions separated by 15 s were performed with each load (total of 20 repetitions per session). The rest period between the different loads was 4 min. The sequence of the exercises and loads was randomized across subjects, but the same order was kept for individual subjects in the two testing sessions. Four common loads were used in the BP and BPT exercises: minimum load (29 kg; mass of the unloaded Smith machine barbell), maximum load (70%1RM: 59.8 ± 15.0 kg), and two intermediate loads that were equidistantly distributed between the minimum and maximum loads (39.4 ± 5.5 and 49.9 ± 10.0 kg). Subjects also performed repetitions during the free-weight BP against two very light loads (one and eight kg). A wooden bar was used for the one kg loading condition, a thin metal bar for the eight kg loading condition, and a traditional Olympic barbell coupled with weight discs were used for the four heavier loading conditions.

The mean propulsive values of force and velocity of each repetition were computed with a linear position transducer (Real Power Pro Globus, Codogne, Italy) that was attached perpendicularly to the barbell. The linear position transducer sampled the displacement-time data at a frequency of 1,000 Hz. Two consecutive derivations of the displacement-time data provided the barbell's velocity and acceleration, respectively. Force was then calculated as the product of the lifted mass and total acceleration (gravity + acceleration of the barbell). Following the guidelines of *Rahmani et al. (2018)*, the mass of the arms (10% of body mass for the two upper limbs according to Winter's table (*Winter, 1990*) and the friction force of the Smith machine determined during a freefall test) were considered for force computations. The propulsive phase was defined as the portion of the concentric phase during which the measured acceleration was greater than acceleration due to gravity (i.e., bar acceleration $> -9.81$ m·s$^{-2}$) (*García-Ramos et al., 2016*). Three different combination of loads were used to determine the F–V relationship: (I) *6-loads free method* (six loads performed during the free-weight BP exercise), (II) *4-loads free method* (four heavy loads performed during the free-weight BP exercise; that is, the very light loads (one and eight kg) were excluded), and (III) *4-loads Smith method* (four loads performed during the BPT exercise in a Smith machine). Only the data of the second testing session were used to address the aims 1 (linearity of the F–V relationship) and 3 (concurrent validity of the F–V relationship parameters), while the data of the second and third testing sessions were used to address aim 2 (reliability of the F–V relationship parameters).

## Statistical analyses

Descriptive data are presented as means and standard deviation. The Pearson's correlation coefficients ($r$) are presented through the median values and range. A one-way repeated

measures ANOVA was applied on the Fisher's $Z$-transformed $r$ coefficients to compare the strength of the individual F–V relationships between the 6-loads free, 4-loads free, and 4-loads Smith methods. The reliability was assessed through paired samples $t$ tests, the Cohen's $d$ effect size (ES), the standard error of measurement (SEM), the coefficient of variation (CV), the intraclass correlation coefficient (ICC), and the corresponding 90% confidence intervals. The ratio between two CVs was used to compare the reliability of the F–V relationship parameters between the 6-loads free, 4-loads free, and 4-loads Smith methods. The smallest important ratio of CVs was considered to be higher than 1.15 (*Fulton et al., 2009*). Paired samples $t$ tests and $r$ coefficients were used to explore the concurrent validity of the F–V relationship parameters obtained from the 6-loads free and 4-loads free methods with respect to the 4-loads Smith method. Qualitative interpretations of the $r$ coefficients as defined by *Hopkins et al. (2009)* (0.00–0.09 trivial; 0.10–0.29 small; 0.30–0.49 moderate; 0.50–0.69 large; 0.70–0.89 very large; 0.90–0.99 nearly perfect; 1.00 perfect) are provided for all significant correlations. The reliability analyses were performed by means of a custom spreadsheet (*Hopkins, 2000*), while other statistical analyses were performed using the software package SPSS (IBM SPSS version 22.0; Chicago, IL, USA). Alpha was set at 0.05.

## RESULTS

### Linearity of the F–V relationship

All individual F–V relationships were highly linear ($r \geq 0.87$). The strength of the individual F–V relationships did not significantly differ ($F = 1.66$, $p = 0.204$) between the 6-loads free ($r = 0.99$ (0.98, 1.00)), 4-loads free ($r = 0.99$ (0.90, 1.00)) and 4-loads Smith ($r = 0.99$ (0.87, 1.00)) methods. Figure 1 depicts the F–V relationships obtained by a representative subject through the three different methods.

### Reliability of the F–V relationship parameters

Comprehensive information regarding the reliability of the F–V relationship parameters is presented in Table 1. The comparison of the CVs suggests that the three methods analyzed in the present study can be ranked from the most to the least reliable as follows: 6-loads free > 4-loads Smith > 4-loads free (Fig. 2). The higher reliability of the 6-loads free method was especially pronounced for F–V slope ($CV_{ratio} \geq 1.85$) and $V_0$ ($CV_{ratio} \geq 1.49$) parameters, while the lowest difference in reliability was observed for $F_0$ ($CV_{ratio} \leq 1.27$). The reliability of the individual points used for the F–V modeling is presented in Table 2.

### Concurrent validity of the F–V relationship parameters

The concurrent validity of the 6-loads free and 4-loads free methods was dependent on the F–V relationship parameter considered (Fig. 3). Regardless of the method, nearly perfect correlations were observed for $F_0$ and $P_{max}$ ($r \geq 0.89$) and small–moderate correlations were observed for $V_0$ ($r \leq 0.37$). The magnitude of the correlation for the F–V slope was very large for the 6-loads free method ($r = 0.82$) and large for the 4-loads free method ($r = 0.66$).

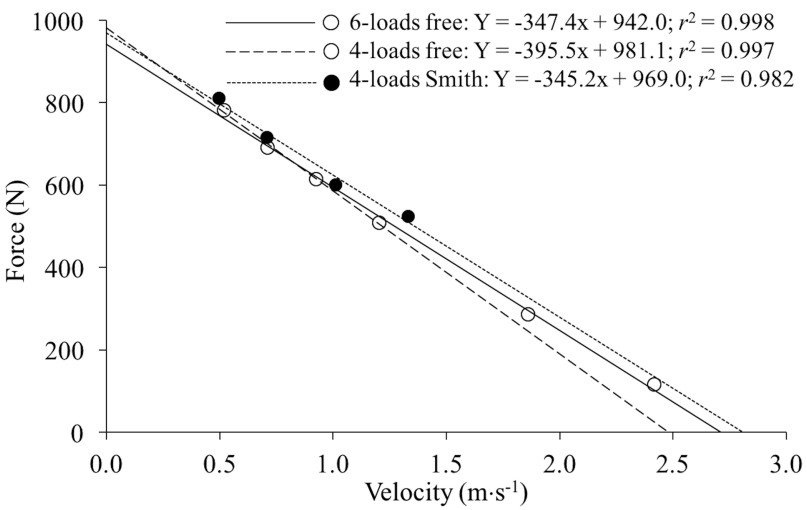

**Figure 1 Individual force–velocity relationships of a representative subject obtained from the 6-loads free (empty dots and straight line), 4-loads free (empty dots and long dashed line) and 4-loads Smith (filled dots and short dashed line) methods.** The regression equations and the Pearson's coefficient of determination ($r^2$) are presented. Note that the points corresponding to the four heavy loads were the same for the 6-loads free and 4-loads free methods, while only the 6-loads method also considered the data of the two very light loads (one and eight kg).

**Table 1 Reliability of the force–velocity relationship parameters obtained from the 6-loads free, 4-loads free, and 4-loads Smith methods.**

| Method | Parameter | Session 1 | Session 2 | $p$ | ES | SEM | CV (90% CI) | ICC (90% CI) |
|---|---|---|---|---|---|---|---|---|
| 6-loads free | $F_0$ (N) | 850 (176) | 848 (195) | 0.803 | −0.01 | 30.7 | 3.62 (2.86, 5.01) | 0.98 (0.95, 0.99) |
| | $V_0$ (m·s⁻¹) | 2.64 (0.22) | 2.66 (0.16) | 0.654 | 0.10 | 0.13 | 5.05 (3.99, 7.00) | 0.53 (0.19, 0.76) |
| | $a$ (N·s·m⁻¹) | 322 (63) | 320 (77) | 0.810 | −0.02 | 21.6 | 6.72 (5.31, 9.30) | 0.92 (0.82, 0.96) |
| | $P_{max}$ (W) | 565 (141) | 563 (134) | 0.827 | −0.01 | 27.5 | 4.88 (3.85, 6.75) | 0.96 (0.92, 0.98) |
| 4-loads free | $F_0$ (N) | 876 (198) | 873 (185) | 0.796 | −0.02 | 33.3 | 3.81 (3.00, 5.27) | 0.97 (0.94, 0.99) |
| | $V_0$ (m·s⁻¹) | 2.55 (0.44) | 2.52 (0.48) | 0.812 | −0.05 | 0.27 | 10.4 (8.24, 14.5) | 0.69 (0.43, 0.85) |
| | $a$ (N·s·m⁻¹) | 356 (108) | 356 (88) | 0.983 | 0.00 | 51.7 | 14.5 (11.5, 20.1) | 0.75 (0.51, 0.88) |
| | $P_{max}$ (W) | 552 (135) | 550 (155) | 0.898 | −0.02 | 52.5 | 9.53 (7.52, 13.2) | 0.88 (0.76, 0.95) |
| 4-loads Smith | $F_0$ (N) | 900 (198) | 925 (205) | 0.073 | 0.13 | 41.8 | 4.59 (3.62, 6.35) | 0.96 (0.92, 0.98) |
| | $V_0$ (m·s⁻¹) | 2.83 (0.30) | 2.72 (0.39) | 0.112 | −0.33 | 0.21 | 7.52 (5.94, 10.4) | 0.66 (0.37, 0.83) |
| | $a$ (N·s·m⁻¹) | 324 (88) | 351 (104) | 0.063 | 0.28 | 41.9 | 12.4 (9.81, 17.2) | 0.83 (0.66, 0.92) |
| | $P_{max}$ (W) | 630 (122) | 620 (125) | 0.361 | −0.08 | 31.8 | 5.10 (4.02, 7.06) | 0.94 (0.87, 0.97) |

**Note:**
$F_0$, maximum force; $V_0$, maximum velocity; $a$, force–velocity slope; $P_{max}$, maximum power; SEM, standard error of measurement; CV, coefficient of variation; ICC, intraclass correlation coefficient; 90% CI, 90% confidence interval.

## DISCUSSION

The main finding of the present study was that the BP performed with a free-weight barbell, which is able to provide experimental points closer to the velocity-intercept, can be used to evaluate the F–V relationship with a high accuracy. The specific findings revealed that (I) the linearity of the F–V relationship was very high and comparable for the three F–V methods, (II) the 6-loads method provided the most reliable F–V relationship

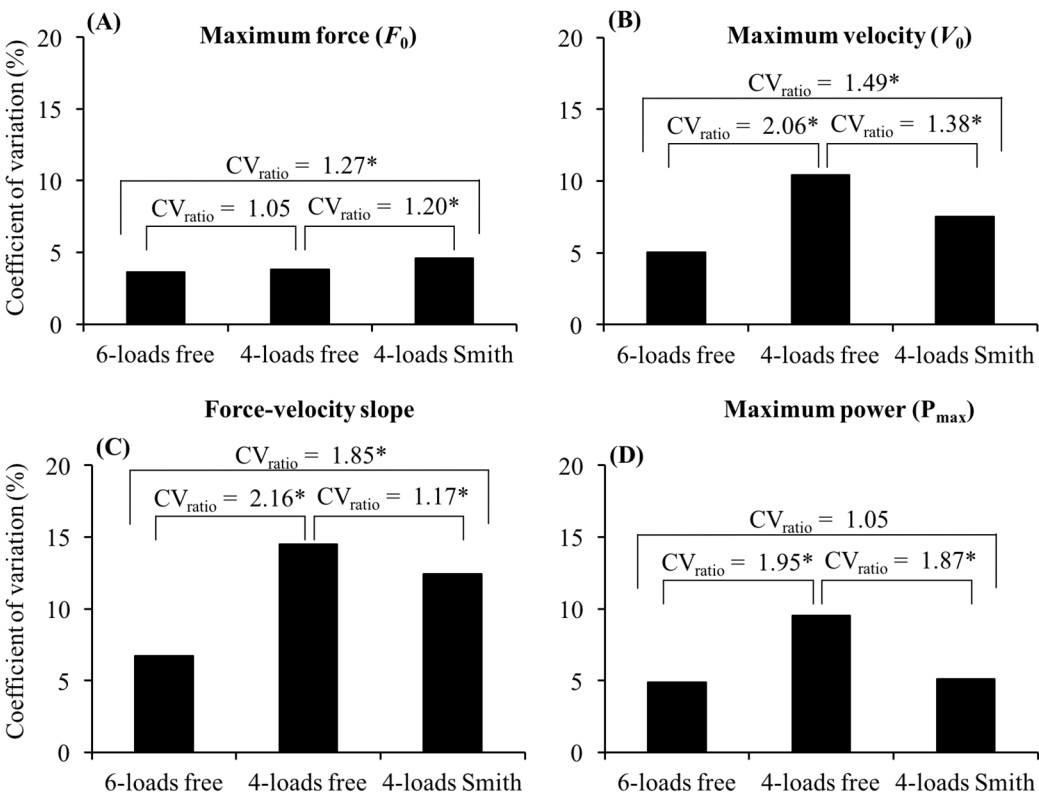

**Figure 2 Comparison of the coefficient of variation (CV) obtained for (A) maximum force, (B) maximum velocity, (C) force–velocity slope, and (D) maximum power between the 6-loads free, 4-loads free and 4-loads Smith methods.** The $CV_{ratio}$ is calculated as higher CV/lower CV. $*CV_{ratio} > 1.15$.

(especially regarding $V_0$ and F–V slope parameters), and (III) the 6-loads free and 4-loads free methods showed a very high validity respect to the 4-loads Smith method for $F_0$ and $P_{max}$, a moderate validity for the F–V slope, and a low validity for $V_0$.

The main novelty of the present study was the inclusion of two very light loads of one and eight kg for modeling the F–V relationship during the BP exercise. The free-weight BP exercise was chosen because it allows to use lower loading conditions than the most commonly used BPT exercise performed in a Smith machine. A potential advantage of using very light loading conditions is the reduction of the extrapolation needed to reach the velocity-intercept from the experimental data (*García-Ramos & Jaric, 2018*). Note that the accuracy in the determination of the F–V relationship seems to be enhanced when the experimental points are located close to the axis intercepts (*Pérez-Castilla et al., 2018*). However, a lower reliability for the 6-loads free method could also be expected since previous studies have reported that velocity outputs are obtained with a lower reliability during the BP compared to the BPT exercise (*García-Ramos et al., 2015*). In addition, the reliability of the 6-loads free method could also be deteriorated because the measurement accuracy of linear position transducers could be compromised during free-weight exercises that include horizontal movements of the barbell compared to the exercises performed in a Smith machine. In general, the results of the present study

**Table 2 Reliability of force and velocity outputs attained at the individual loads used for the force–velocity modeling.**

| Exercise | Variable | Load | Session 1 | Session 2 | p | ES | SEM | CV (90% CI) | ICC (90% CI) |
|---|---|---|---|---|---|---|---|---|---|
| BP | Force (N) | 1 | 117 (13) | 117 (13) | 0.911 | 0.01 | 2.9 | 2.45 (1.93, 3.39) | 0.96 (0.91, 0.98) |
| | | 2 | 277 (35) | 276 (30) | 0.846 | −0.01 | 7.4 | 2.68 (2.12, 3.71) | 0.95 (0.90, 0.98) |
| | | 3 | 484 (56) | 482 (61) | 0.811 | −0.02 | 16.0 | 3.32 (2.62, 4.60) | 0.93 (0.86, 0.97) |
| | | 4 | 558 (89) | 556 (97) | 0.706 | −0.03 | 19.5 | 3.50 (2.76, 4.85) | 0.96 (0.92, 0.98) |
| | | 5 | 627 (127) | 626 (124) | 0.933 | −0.01 | 26.7 | 4.26 (3.36, 5.89) | 0.96 (0.91, 0.98) |
| | | 6 | 692 (149) | 695 (152) | 0.693 | 0.02 | 22.2 | 3.20 (2.53, 4.44) | 0.98 (0.96, 0.99) |
| | Velocity (m·s$^{-1}$) | 1 | 2.27 (0.25) | 2.26 (0.16) | 0.795 | −0.04 | 0.11 | 4.75 (3.75, 6.58) | 0.76 (0.53, 0.88) |
| | | 2 | 1.77 (0.18) | 1.79 (0.15) | 0.548 | 0.10 | 0.09 | 4.81 (3.80, 6.66) | 0.76 (0.54, 0.88) |
| | | 3 | 1.08 (0.21) | 1.08 (0.24) | 0.963 | 0.00 | 0.07 | 6.09 (4.81, 8.43) | 0.93 (0.84, 0.97) |
| | | 4 | 0.86 (0.15) | 0.87 (0.14) | 0.767 | 0.05 | 0.07 | 8.64 (6.83, 11.97) | 0.76 (0.54, 0.89) |
| | | 5 | 0.70 (0.10) | 0.67 (0.11) | 0.133 | −0.22 | 0.05 | 6.83 (5.40, 9.46) | 0.83 (0.66, 0.92) |
| | | 6 | 0.52 (0.12) | 0.51 (0.08) | 0.607 | −0.11 | 0.07 | 13.40 (10.58, 18.56) | 0.58 (0.25, 0.78) |
| BPT | Force (N) | 1 | 503 (46) | 499 (49) | 0.394 | −0.07 | 12.2 | 2.45 (1.93, 3.39) | 0.94 (0.88, 0.97) |
| | | 2 | 578 (92) | 575 (89) | 0.556 | −0.03 | 14.9 | 2.58 (2.04, 3.57) | 0.98 (0.95, 0.99) |
| | | 3 | 645 (118) | 655 (120) | 0.161 | 0.08 | 20.3 | 3.13 (2.47, 4.33) | 0.97 (0.94, 0.99) |
| | | 4 | 713 (148) | 722 (155) | 0.194 | 0.06 | 19.9 | 2.77 (2.19, 3.83) | 0.98 (0.97, 0.99) |
| | Velocity (m·s$^{-1}$) | 1 | 1.19 (0.22) | 1.18 (0.21) | 0.968 | −0.01 | 0.10 | 8.32 (6.57, 11.51) | 0.81 (0.62, 0.91) |
| | | 2 | 0.95 (0.14) | 0.96 (0.13) | 0.714 | 0.06 | 0.07 | 7.68 (6.06, 10.63) | 0.74 (0.50, 0.87) |
| | | 3 | 0.76 (0.12) | 0.77 (0.10) | 0.873 | 0.04 | 0.08 | 10.33 (8.16, 14.30) | 0.52 (0.18, 0.75) |
| | | 4 | 0.58 (0.11) | 0.59 (0.08) | 0.635 | 0.11 | 0.06 | 10.98 (8.67, 15.21) | 0.55 (0.22, 0.77) |

**Note:**

BP, bench press performed with a free-weight barbell; BPT, bench press throw performed in a Smith machine; SEM, standard error of measurement; CV, coefficient of variation; ICC, intraclass correlation coefficient; 90% CI, 90% confidence interval.

confirmed our initial hypotheses, highlighting that a more accurate F–V relationship can be obtained from the 6-loads free method than from the standard 4-loads Smith method.

The analysis of the F–V relationships obtained from the 6-loads method allowed us to confirm our first hypothesis; that is, the F–V relationship remained highly linear when modeled using experimental points very close to the velocity-intercept. Similarly, *Riviere et al. (2017)* revealed during the squat jump exercise that the 1RM can be considered as a point of the F–V relationship since the goodness of fit of the individual F–V relationships were not significantly different with or without including the point of the 1RM condition. Therefore, while *Riviere et al. (2017)* showed that the F–V relationship remained linear when an experimental point very close to the force-intercept (1RM load) was modeled, in the present study we found similar results when two points very close to the velocity-intercept (one and eight kg) were considered. In general, these results further support the high linearity of the F–V relationship that has been described for several exercises (e.g., vertical jumps, running, cycling, bench pull, BPT, leg extension, etc.) (*Cross et al., 2018*; *García-Ramos et al., 2016*, *2018b*; *Iglesias-Soler et al., 2018*; *Lemaire et al., 2014*; *Zivkovic et al., 2017*).

The second hypothesis of the study was also confirmed since the 6-loads free method resulted to be the most reliable procedure. As expected, the increased reliability of the 6-loads free method was more pronounced for $V_0$ than for $F_0$. These results are in line

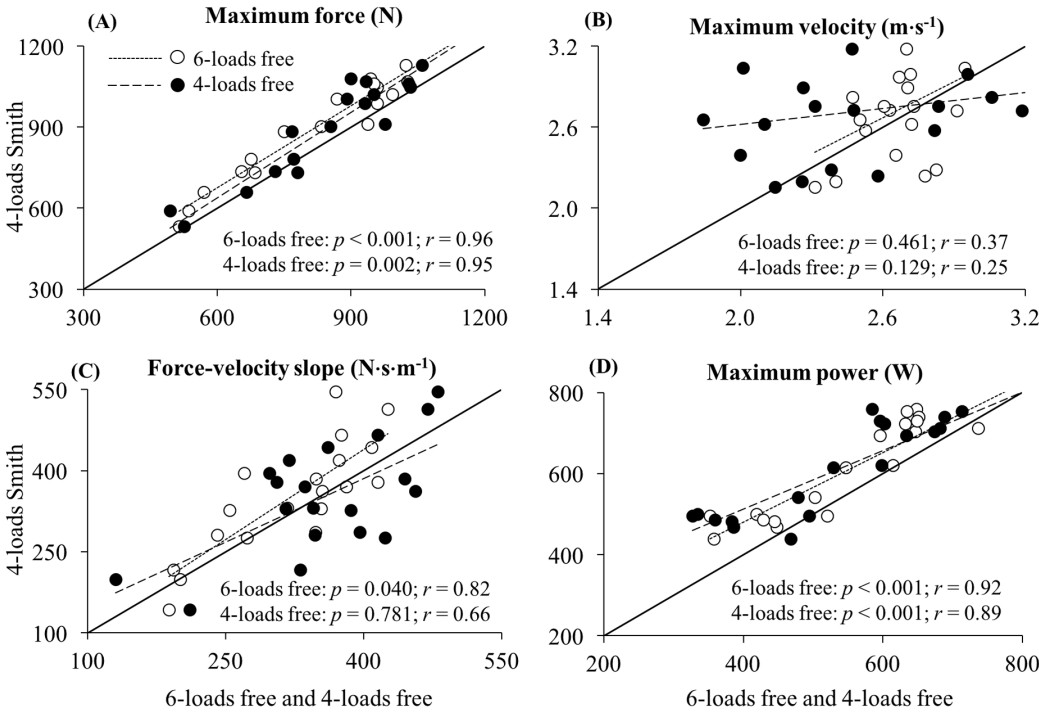

**Figure 3 Concurrent validity of (A) maximum force, (B) maximum velocity, (C) force–velocity slope, and (D) maximum power parameters.** The force–velocity relationship parameters obtained from the 6-loads free (empty dots and short dashed line) and 4-loads free (filled dots and long dashed line) methods were compared against the standard 4-loads Smith method. The identity line (straight line) is depicted. $p$, $p$-value obtained from paired samples $t$-tests; $r$, Pearson's correlation coefficient.

with the study of *Pérez-Castilla et al. (2018)* who reported that force-biased data points (50–70%1RM) provide $F_0$ with higher reliability, while velocity-biased data points (20–40%1RM) are preferable to gain precision in the determination of $V_0$. Since the experimental points used in the present study for the 4-loads Smith method were generally force-biased (see Fig. 1), it is reasonable that the reliability of $F_0$ was higher than for $V_0$. Our third hypothesis was rejected since the 6-loads free method did not provide a higher concurrent validity than the 4-loads free method. Both methods showed a very high concurrent validity for $F_0$ and $P_{max}$, a moderate validity for the F–V slope, and a low validity for $V_0$. The lower validity of $V_0$ and the F–V slope could be partially explained by their lower reliability, while the lower between-subjects variability in the values of $V_0$ compared to $F_0$ could have also reduced the magnitude of the $r$ coefficients for $V_0$. In any case, the very high correlations observed for $F_0$ and $P_{max}$ support the use of the free-weight BP exercise for testing the F–V relationship. Based on these results, a general recommendation for accurately determining the F–V relationship is to have one point as close as possible to the force-intercept and another point as close as possible to the velocity-intercept. However, it is important to note that the optimal magnitude of the extreme points of the F–V relationship should be carefully determined for each exercise since very extreme points could present a low reliability (e.g., velocities lower than 110 revolutions per minute in the leg cycle ergometer exercise are not recommended;

(*García-Ramos et al., 2018a*) and, consequently, their use could compromise the accuracy of the F–V relationship.

The main limitation of the present study is that the minimum load of the Smith machine was 29 kg, which represents a heavier load than the minimum load used in previous studies for determining the F–V relationship during the BPT exercise in a Smith machine (≈ 17–20 kg) (*García-Ramos et al., 2016*; *Pérez-Castilla et al., 2018*; *Rahmani et al., 2018*; *Sreckovic et al., 2015*). In this regard, it is plausible that the use of a lighter Smith machine barbell could have promoted lower differences in reliability between the 6-loads free and 4-loads Smith methods. Therefore, although the use of a very light Smith machine barbell could be recommended for assessing the F–V relationship during the BPT exercise, sport practitioners should be aware that the F–V relationship could also be accurately determined during the free-weight BP exercise. Future studies should try to identify which is the optimal range of loads that are able to maximize the accuracy in the determination of the F–V relationship during the BP exercise as well as during other basic exercises (e.g., vertical jumps, cycling, sprinting, etc.).

## CONCLUSIONS

The F–V relationship remained highly linear when experimental points close to the velocity-intercept were used for the F–V modeling (i.e., 6-loads free method). The 6-loads free method generally provided the F–V relationship parameters with higher reliability than the 4-loads free and 4-loads Smith methods. While a very high concurrent validity of the 6-loads free and 4-loads free methods with respect to the 4-loads Smith method was observed for $F_0$ and $P_{\max}$, the validity was considerably lower for the F–V slope and especially for $V_0$. Taken together, these results emphasize the importance of including experimental points close to the axis intercepts to enhance the accuracy of the F–V relationship. Therefore, the routine testing of the F–V relationship of upper-body muscles through the BP exercise should include trials with very light loading conditions to enhance the reliability of the F–V relationship. However, we recommend avoiding extreme points when they have a low reliability because they could compromise the accuracy of the F–V relationship.

### Funding
The authors received no funding for this work.

### Competing Interests
The authors declare that they have no competing interests.

### Author Contributions
- Jesualdo Cuevas-Aburto conceived and designed the experiments, performed the experiments, analyzed the data, contributed reagents/materials/analysis tools, prepared figures and/or tables, authored or reviewed drafts of the paper, approved the final draft.

- David Ulloa-Díaz conceived and designed the experiments, performed the experiments, contributed reagents/materials/analysis tools, authored or reviewed drafts of the paper, approved the final draft.
- Paola Barboza-González conceived and designed the experiments, performed the experiments, contributed reagents/materials/analysis tools, authored or reviewed drafts of the paper, approved the final draft.
- Luis Javier Chirosa-Ríos conceived and designed the experiments, contributed reagents/materials/analysis tools, authored or reviewed drafts of the paper, approved the final draft.
- Amador García-Ramos conceived and designed the experiments, performed the experiments, analyzed the data, contributed reagents/materials/analysis tools, prepared figures and/or tables, authored or reviewed drafts of the paper, approved the final draft.

### Human Ethics

The following information was supplied relating to ethical approvals (i.e., approving body and any reference numbers):

The study protocol was approved by the Institutional Review Board of the University of Granada (no: 491/CEIH/2018).

### Data Availability

The raw measurements of force and velocity used to determine the F–V relationships during both testing sessions are available in a Supplemental File.

### Supplemental Information

Supplemental information for this article can be found online at http://dx.doi.org/10.7717/peerj.5835#supplemental-information.

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
