# Peer review of "The addition of very light loads into the routine testing of the bench press increases the reliability of the force–velocity relationship"

_PeerJ, doi:10.7717/peerj.5835_

## Round 0.1 · original submission · Major Revisions

Both of the reviewers and I see many positive aspects of your study and the written manuscript and I feel that the constructive criticisms described by two reviewers will further elevate this paper to a very high standard that contribute substantially to the literature and strength and conditioning practice.

·

Basic reporting

No Comment

Experimental design

The experiment design of this study is of a very high quality and the authors should be commended. Well done.

Just to clarify (and to help the reader so that these protocols can be replicated in the future)
I am still a little unclear whether the lighter loads used (e.g. 1 and 8kg) in the traditional bench press were additional to the bar weight?
I am also a little unsure whether a traditional Olympic barbell in the traditional bench press exercise was used? Can this be clarified?
I believe that if the 1 and 8kg are being added to the bar it could be seen as misleading to not state actual total load. This issue may be a due to a small lack of clarity in the text which is only a minor fix.

Validity of the findings

No comment

Additional comments

Line 93: A very small thing but this sentence could be tidied up a little. Currently it is quite difficult to understand (“velocity-intercept is responsible that we still”). I think a word or two might be missing

Line 100: perhaps rearranging the words from “has been commonly used” to “…since a narrow range of loads have commonly been used…” might make the sentence read a little better?

Line 118: I think the words “in the present study” can be removed

Sentence starting on line 156: It would be appreciated if the sentence starting “The F-V relationship….” Could be made a little bit clearer. There are a few words missing and it starts to get quite unclear towards the second half of the sentence

Line 162: “…to address the aim 2” The grammar here isn’t quite right. You could remove the “the” so it reads “….to address aim 2”.

Line 171: “3” Should be “Three” as it’s the start of the sentence

Line 175: I think at the end of the sentence starting line 175 (i.e. concentric only BP) a reference is needed to help direct the reader if they are interested in why this movement was selected. I’m sure the authors will know a good reference that can fit there….

Section “Determination of the 1RM”.
I’m a little bit confused (and this is an easy fix) but was the traditional and BPT using the same loads? Or did they test maximal 1RM on both exercises? If different, how did the authors know when the BPT was no longer a BPT? By clarifying this section a little the reader will be able to understand how to do this testing with their own populations.

Line 198: the authors have referred to the exercises as the traditional and ballistic BP but due to it previously being called the BPT it may confuse the reader (I had to read it a couple of times to make sure that they were the same exercise). Perhaps the same terminology could be used just to clarify for the reader?

Line 243: a bit nit picky but the authors have used the word “strong” to describe the F-V relationship. Should this be “nearly perfect” (for example), as these are the terms outlined in the statistical analysis section?

Line 337: a second bracket is needed after “2018)”

Reviewer 2 ·

Basic reporting

No comment - see general comments below

Experimental design

No comment - see general comments below

Validity of the findings

No comment - see general comments below

Additional comments

The aims of the study entitled « The addition of very light loads into routine testing of the bench press increases the reliability of the force-velocity relationship » were:
I) Test whether the evaluation of the force-velocity (F-v) relationship in bench press throw (BPT) using a machine-guided bar, which is the usual testing machine, could be replaced by a traditional bench press (BP) evaluation without guided bar.
II) Test whether the addition of very light loads to the F-v during traditional BP testing could improve the reliability of extrapolated parameters, notably V0.
III) Test whether the F-v relationship is still linear on the velocity side, which has not been experimentally explored in bench press of movement.
These goals were motivated because of technical issues specifically due to heavy machinery that lead the F-v relationship to be explored in a shorten range, notably closer to the force side. In consequence, the quality of the testing, that is, the reliability of associated parameters to define neuromuscular capacities, could be negatively influenced.
In general, I commend the good work done by the author and the excellent initiative toward improving strength & conditioning methods. Nevertheless, I believe that there is still a room for improving the manuscript. Therefore, the remaining text represents the list of my general and specific comments and suggestions, although, I will understand, with logical and justified answers, if author decide not to address all of them in the revised version.

GENERAL COMMENT:
For each section of the manuscript, the content is sufficient to well understand the manuscript and its goals. I also appreciate the effort of authors for the overall clear-written sentences. However, some additional information is still needed to facilitate the reading while other sentences should be deleted because there are not enough informative and make the reading difficult. Also, although I am convinced by the pertinence of the study, I would strongly suggest the authors to reorganize the order of presentation of the goals as suggested in the paragraphs above. I am convinced that would improve significantly the understanding and help readers to identify more easily the three main goals.
Please, feel all my comments as objective suggestions with no negative judgements and with the only idea to improve the understanding for all of your excellent manuscript.

SPECIFIC COMMENTS:

INTRODUCTION
Lines 59 – 61: Since the force-velocity relationship is related to the neuromuscular system force capabilities, I am concern about the role of osteoarticular system.
Could you explain and justify with appropriated references how the latter independently influences a force-velocity relationship?
Moreover, the power-velocity relationship is also well used to explore the function of the neuromuscular. This relationship should be presented at these lines and added consequently in the following sentences of the manuscript.
Lines 61 – 64: The main ability of muscle is to transform chemical energy into a mechanical force output. The movement velocity is a consequence and the power is a mathematical construction rather than “muscular productions”. Also, Pmax is not theoretical, it is surely attaint in numbers of movements such as squat jump or cycling.
I suggest authors to clarify this sentence and to consider the following suggestion: “The modelling of the F-V-P relationships in single- and multi-joint tasks, allows to determine mechanical limits of the neuromuscular system being theoretical maximal force capacity at null velocity (F0), the theoretical maximal velocity capacity until which the system is still able to produce force (V0) and the maximal power capacity (Pmax).
At line 64, To avoid any confusion, I would replace the word “property” by the word “characteristic” since the word “property” is also use to qualify muscles.
Lines 64 – 71: From my point of view, both sentences express the same idea. I would recommend to synthetize them into one sentence to shorten the reading as this idea is not the main point of the study.
Also at lines 67 and 68, I would suppress the brackets to define F0 and V0, which as already been defined in the above sentences.
Lines 80: If possible, I would add a small paragraph about how F-v relationship testing is usual done: number of loads, measurements (in laboratory and on the field), apparatus (e.g. force plate, phone). I am convinced that would help readers to understand the limitations of these procedures.
Lines 82 – 83: As this is the main issue, I recommend to introduce here the fact that, both, a wider range and the number of point as a direct impact on the reliability of the F-v relationship.
Lines 84 – 88: I would replace the word “unfortunately” by the word “However” since the latter does not include personal feelings.
Also at line 85, I recommend to clarify the fact that the range of loads is limited in some exercise because, on the field, technical supports to extend the range are not often easily available in contradictory to testing conducted in laboratory.
At line 88, could you justify how the smith machine is considered as a standard procedure?
The justifications should be added in a short following sentence or associated to the previous sentence. I propose to replace the word for “the most common and often used method”.
Lines 92 – 96: The sentence should be clarified and authors could, if agreed, followed this suggestion “Since the minimum load imposed by the unloaded Smith machine system is approximately 17-20 kg, the resulting experimental points to determine the F-V relationship during the BPT exercise are generally closer to the force-intercept than to the velocity-intercept (i.e. force-biased).”
Lines 96 – 102: I would suppress these sentences because they address unclearly the main goals without being very specific. I recommend to replace this paragraph for quantitative information about how far the experimental points obtain with the smith machine is from V0. I would also add quantitative information about previous studies (if any) that have used very light load during BPT and measured force and velocity data or computed a force-velocity relationship. This would inform readers about the gap to fill until V0 is reached.
Lines 104 – 117: As previously suggested, I would recommend the authors to reorganize the order of the goals and to write specific sentences for each goal as readers can identify them easily.
At lines 104 to 106, I am concerned about the justification of the F-v relationship linearity testing. The authors should provide appropriate references to convinced readers that the linearity is still challenged.
At lines 115 to 117, the use of the words “to approximate” is not clear can you clarify this sentence?
Lines 119-138, I strongly advice authors to shorten this paragraph by suppressing lines 119 to 125 and directly introducing the main goal of the study. Since the paper is very short, there is no need to add a sum up, even less so the sum up is quite as long as the introduction. I am convinced that would help the understanding of the study. I also recommend to dissociate, in two separated paragraph, the aims of the study from the hypotheses with, both, short and precise sentences.

METHOD
Lines 142 – 144: I would shift the results about one repetition maximum (1RM) in BP toward the result section.
Lines 153 – 155: I would suppress this sentence, because, from my point of view, it does not help the understanding of the study design.
Lines 157 – 159: This sentence is not in line with a following sentence in the same section, which is: “The testing protocol of both testing sessions consisted of the traditional BP performed against six loading conditions and the ballistic BPT performed against four loading conditions.”
Actually, if I well understood, subjects performed two testing sessions: a F-v relationship testing in BP under 6 loaded conditions and a F-v relationship testing in Smith machine under 4 loaded conditions. Using 4 or 6 loads to compute different F-v relationships in BP is rather an analysis strategy. I would recommend here to suppress the bracket and just to introduce the F-v relationships determination for two movements in BP and BPT rather than with three methods.
Lines 160 – 169: This paragraph should be suppressed. From my point of view, it is confusing to address these specific points at the beginning of the method section when readers have few information about the whole protocol. Also, all of these information is almost all addressed in a next paragraph.
Lines 173 – 175: Could you be more specific about the joint warmed-up during the joint mobility exercises and the details of the jogging (duration and running velocity)?
Lines 175 – 178: Could you be more specific about the steps of increasing the load during the 1RM determination? Notably, how did the steps of increase 10, 5, 2, or 1 kg was chosen and what was the objective criteria to adapt the step.
Lines 178 – 180: I don’t understand why two repetitions were performed since this test is completed as the subject is able to perform just one repetition? Could you clarify why this was done?
Lines 186 – 188: I also don’t understand why subjects were to ask perform the repetition as quickly as possible. The aim here was to assess if the subject is able to lift the load rather than to throw as high as possible. Could you clarify why this was done?
Lines 188: The brackets are not clear; I would suggest authors to suppress it as it has been explained before.
Lines 188 – 192: These sentences should shift towards the next sub-section.
Lines 197 – 199: Could you explain why they performed BPT with the unloaded smith machine?
Lines 207: The 29kg of the unloaded smith machine is not similar as the data presented in the introduction: “The experimental points used […] the minimum load applied is approximately 17-20 kg (mass of the unloaded Smith machine) […].”. Could you clarify this point?
Lines 213 – 215: I would suggest the authors to precise that force and velocity data are computed from a linear transducer rather recorded. Also, the method to compute the force and velocity data from the linear transducer should be presented as well as the signal treatment (filtering or average moving window).
Also, while I respect the authors decision to focus on the propulsive phase, I would like to highlight that depending of the loading condition, the propulsive phase during BP do not consider the same push-off distance. The averaged force and velocity data come from more or less complete part of the movement (the lower the load, the shorter the push-off distance). In consequence, the data does not represent the subject capability to produce force in the same condition. Since the force-velocity relationship should represent so, could you justify how the use of the propulsive phase could be interpreted?
Lines 223 – 226: These sentences could be suppressed.
Lines 235: Could you rather express the standard error of measurement relatively to the mean of both sample? The SEM in absolute value does not fully inform about the error and can be bias depending on the magnitude of the dependent variable, as suggested By Hopkins in 2001.

RESULTS
Lines 253 – 255: I suggest the authors to change this sentence and only talk about individual results, since a mean F-v relationship across a sample is difficult to interpret; There is actually no interest to determine a mean F-v relationship across subjects except when team work is performed such as in rowing.
This suggestion is also applicable to the figure 1. I would prefer a figure with a typical subject rather than a figure that inform about the mean variability between subject which is not informative regarding the aims of the study.
Lines 273 – 278: The validity of BP with 6 loads against smith machine with 4 loads included different number of load and different range. I’m afraid that the effect of range and number of point may influence the potential link between BP and smith machine. This limitation is over come for the comparison of BP with 6 loads to smith machine with 4 loads. Can the authors justified why they decided to compare a 6 load methods to a 4 loads method to test the validity?
Overall, I strongly suggest the authors to add an additional table with mean +/- SD force, velocity and power across subjects for each loading condition in both methods. Firstly, it would help the understanding of the protocol and secondly the data could be used for potential meta-analysis or use as a scientific data base being a solid and quantitative knowledge.
DISCUSSION
Lines 347 – 352: I personally find this discussion very interesting and I think that adding the intra-subject coefficient of variation between the trials of the two light loads could strengthen the discussion and give very interesting information about future studies on the reliability of very high velocity movement.
Lines 283 – 285: I would recommend the authors to suppress this sentence.
Lines 286 – 294: I would reorganize the aims’ order as suggested above.
Lines 291 – 294: This sentence should be shift towards the conclusion section.
Lines 296 – 323: This paragraph is not clear. It is not possible to dissociate if some sentence are statements based on results of this study or other studies or either hypotheses. I suggest the author here to reorganize the paragraph and use results to support their statement which could help to clarify the outcomes of the study.

CONCLUSION
I would suggest the authors to also highlight that despite the fact that extreme point could increase the F-v quality by reducing the extrapolation of parameters of interest, on the over hand, due to the low reliability of these unusual movement, the positive effect could be turn into a negative effect. This point may catch attention about the importance of familiarization before using these extreme conditions into testing routine in order to avoid methodological bias.

REFERENCES
Authors’ references in the text is not conventional, when there are more than 2 authors in publication, only the first author’s name and the formulation “et al., + year of publication” should be cited, otherwise the both authors name should be cited. This comment should be applied to entire the manuscript.
Also, the citation of articles in the reference section are not suitable. I recommend the authors to check the review's guidelines.

---

## Round 0.2 · Minor Revisions

The authors have attended to many of the initial concerns of the reviewers. Please see the remaining comments from reviewer two that need to be attended to prior to this paper be accepted for publication.

·

Basic reporting

I believe that the authors have done an excellent job and should be commended on their reporting. The English was excellent, they have provided a great background, and the tables and figures are of a high quality. Well done.

Experimental design

The research question is clear and the investigation was rigorous. All methods were well described.

Validity of the findings

This is a well designed study that has good impact and is novel.

Additional comments

Well done on this manuscript. It is great to see more research in the velocity based training, force/power area. I look forward to seeing this in print in the near future.

Reviewer 2 ·

Basic reporting

I would like to thank the authors to have some much consideration for both reviewers’ comments and suggestions. The detailed revision of the manuscript is highly appreciated, as well as the honest and justified answer that the authors provided.
I feel that the manuscript is almost ready to be published since few important details need to be still discussed or adjusted.

ABSTRACT
Lines 29 – 30: I would replace a part of this sentence:
“The aim of this study was to examine whether the addition of very light loads for modelling the force-velocity (F-V) relationship during the bench press exercise can increase its linearity […]”

by

“The aim of this study was to examine whether the addition of very light loads for modelling the force-velocity (F-V) relationship during the bench press exercise can confirm its experimental linearity […]”

METHODS
Line 199 – 201: I thank the authors for being more specific about the computation but the way to compute the force need to be more detailed. I’m concerned about the fact that the force computation, if I well understood, is F = Total mass . (gravitational acceleration + barbell acceleration).

My first interrogation: is the total mass comprise the mass of the barbell, the additional mass and the mass of the upper limbs?

Secondly, have you quantified the frictional force of the smith machine? because it should be added to the computation.

My thinking is based on the study of Rahmani et al in 2001 where they detailed the usual computation to estimate the force production during a bench press movement:

(Abderrehmane, R., Fabrice, V., Georges, D., & Lacour, J.-R. (2001). Force / velocity and power / velocity relationships in squat exercise. European Journal Of Applied Physiology And Occupational Physiology, 84, 227–232. https://doi.org/10.1007/PL00007956)

The computations being F = (mul + mb) . ab + (mul + mb) . g + Ff

With mul, the masse of the upper limbs, mb the mass of the barbell, ab, the acceleration of the barbell, g the gravitational acceleration and Ff, the frictional force.

The authors should confirmed their computation or, unfortunately, re-analyzed the data to compute the force production :

1) To get the true value because such data could be used later in other study analysis

2) because, although excluding the mass of the upper limb during the heavy loading condition could be negligible, during the very-light load conditions, the mass of the upper limb would be substantially higher than the mass of the barbell, and in turn, influencing greatly the force production.

Experimental design

See commentary above

Validity of the findings

See commentary above

Additional comments

See commentary above

---

## Round 0.3 · accepted · Accept

We thank the authors for their hard work and am happy to state that the paper is now accepted for publication.

Reviewer 2 ·

Basic reporting

See commentary below

Experimental design

See commentary below

Validity of the findings

See commentary below

Additional comments

I would like to thank again the authors to have some much consideration in writing their manuscript and getting perfectly involved in collaboration with reviewers !
I encourage them to continue their works in this field !

Enjoy the published paper !